# Optimization of Residual Stress Measurement Conditions for a 2D Method Using X-ray Diffraction and Its Application for Stainless Steel Treated by Laser Cavitation Peening

**DOI:** 10.3390/ma14112772

**Published:** 2021-05-24

**Authors:** Hitoshi Soyama, Chieko Kuji, Tsunemoto Kuriyagawa, Christopher R. Chighizola, Michael R. Hill

**Affiliations:** 1Department of Finemechanics, Tohoku University, Sendai 980-8579, Japan; 2Department of Mechanical Systems Engineering, Tohoku University, Sendai 980-8579, Japan; kuji.shinkou@gmail.com (C.K.); tkuri@tohoku.ac.jp (T.K.); 3Department of Mechanical and Aerospace Engineering, University of California Davis, Davis, CA 95616, USA; crchighizola@ucdavis.edu (C.R.C.); mrhill@ucdavis.edu (M.R.H.)

**Keywords:** residual stress, X-ray diffraction, laser cavitation peening, pulse laser

## Abstract

As the fatigue strength of metallic components may be affected by residual stress variation at small length scales, an evaluation method for studying residual stress at sub-mm scale is needed. The sin^2^*ψ* method using X-ray diffraction (XRD) is a common method to measure residual stress. However, this method has a lower limit on length scale. In the present study, a method using at a 2D XRD detector with *ω*-oscillation is proposed, and the measured residual stress obtained by the 2D method is compared to results obtained from the sin^2^*ψ* method and the slitting method. The results show that the 2D method can evaluate residual stress in areas with a diameter of 0.2 mm or less in a stainless steel with average grain size of 7 μm. The 2D method was further applied to assess residual stress in the stainless steel after treatment by laser cavitation peening (LCP). The diameter of the laser spot used for LCP was about 0.5 mm, and the stainless steel was treated with evenly spaced laser spots at 4 pulses/mm^2^. The 2D method revealed fluctuations of LCP-induced residual stress at sub-mm scale that are consistent with fluctuations in the height of the peened surface.

## 1. Introduction

As residual stress is one of the most important factors related to the fatigue strength of metallic materials [1,2,3,4,5,6,7,8], it is worth measuring the residual stress in local areas subject to fatigue crack nucleation. It is well known that conventional welding causes tensile residual stress near the welded line due to the heat-affected zone (HAZ) [9,10,11,12]. Friction stir welding (FSW) also generates tensile residual stress near the FSW region [13,14,15,16,17,18], as FSW produces stirring and a thermo-mechanically affected zone. Residual stress is one of the key factors for mechanical surface treatments such as shot peening (SP) [19]. Laser peening can also improve fatigue properties by introducing compressive residual stress [20,21,22,23]. As the distribution of the residual stress of conventional welding and the FSWed part drastically changes with distance from the welding line, the residual stress of the laser-peened surface is also distributed with a laser spot size of the mm-order. One of the conventional methods used to evaluate the residual stress is X-ray diffraction. As the size of the measured area using X-ray diffraction is similar to that of the distribution of the residual stress of the welding part and/or the laser-peened surface in sub-millimeter order, it is necessary to improve the accuracy of residual stress measurements by using X-ray diffraction. Note that the most important factor of the stresses measurement accuracy in local area using X-ray diffraction is the number of the grains.

The sin^2^*ψ* method is the most popular method for evaluating the residual stress of polycrystal metals using X-ray diffraction [24], and a 2D method using a two-dimensional detector has been developed [25]. Regarding JSMS standard, 3 × 10^5^ to 6 × 10^5^ grains is required for the sin^2^*ψ* method. Each method is based on its own theory, and each has advantages and disadvantages. For example, in the case of the sin^2^*ψ* method, a simple goniometer is sufficient to evaluate the residual stress. On the other hand, the 2D method can evaluate the 3D stress state, but a highly accurate multi-axis goniometer is needed. The great advantage of using a 2D detector is that the Debye ring can be evaluated by interpolation and extrapolation. Namely, the 2D method using a 2D detector could be used to evaluate residual stress of very small area and/or coarse grain. A 2D method with specimen oscillation by moving the detector in the direction orthogonal to *θ*-direction was proposed to obtain better Debye ring in the reference [26], but the obtained result was not compared with the result obtained using the other method. One way to evaluate residual stress measurements using X-ray diffraction is to compare them with mechanical relaxation method such as a slitting method [27] and/or a hole drilling method [28]. The slitting method is relatively easy to perform, can be performed quickly, and provides excellent repeatability, which makes it very useful for actual laboratory residual stress measurements [27]. As the experimental deviation of the slitting method was smaller than that of the hole drilling method [29], the slitting method was chosen in the present experiment.

As mentioned above, laser peening introduces compressive residual stress and enhances the fatigue properties [20,21,22,23]. Y. Sano et al. measured the residual stress distribution with depth for stainless steel SUS304 and demonstrated an improvement in fatigue strength by laser peening without protective coating [21]. In the case of laser peening with coating [20,22], coating or tapes such as black polymer tape or metal foil is pasted on material to control laser energy absorption and prevent the surface from melting. The distributions of residual stress were precisely measured, but fluctuations due to the laser spot were not observed [30,31]. The distribution of changes in residual stress with depth was precisely measured, but there was no information provided for the distribution on the surface [32,33]. On the other hand, in the case of the numerical simulation, residual stress distributions due to laser spots were clearly observed [6,34,35,36]. Recently, the patterns of residual stress on the surface due to laser spots were also observed [37,38]. G. Xu et al. measured the residual stress of SUS316L by the sin^2^*ψ* method, in which the diameter of the measured area was 2 mm with a 0.5 mm step; the laser spot was 3 × 3 mm^2^; the overlapping rates were 30%, 50% and 70%; and the cyclic pattern of the residual stress was obtained [37]. X. Pan et al. measured the residual stress of Ti6Al4V by the sin^2^*ψ* method, in which the diameter of the measuring area was 2 mm with a 1 mm step, the laser spot was 2.4 mm, the overlapping rate was 40% and a cyclic pattern due to laser spots was not observed in the distribution of residual stress [39]. Using a synchrotron, Y. Sano et al. measured the distribution of the residual stress crossing over a single laser spot with 1D line irradiation by measuring an area of 0.2 mm in diameter; the laser spot was about 1 mm in diameter and the authors reported the tensile residual stress due to the laser spot [38]. It was determined by numerical simulation that the crack propagation was affected by the residual stress distribution due to laser peening [40,41]; therefore, the precise distribution of the residual stress had to be determined. Figure 1 shows the typical pattern of a fractured fatigue specimen of a stainless steel (a) non-peened specimen and (b) laser-peened specimen [23]. As shown in Figure 1b, the fatigue crack of the laser-peened specimen propagated nearly straight compared with that of the non-peened one, in which several cracks were propagated in parallel due to the increase of the hardness near the crack tips by plastic deformation. The distribution of residual stress of laser peened specimen. The distribution of residual stress of the laser peened specimen could be one of the reasons for crack propagation. Considering a previous report [38], the measurement of the residual stress in a submillimeter-order area is required. Thus, a method that can measure residual stress at a submillimeter level using a conventional X-ray diffraction apparatus is needed.

This paper consists of two parts. The first half reveals the optimization of measuring the condition of the 2D method using a two-dimensional detector for residual stress measurements to evaluate the surface modification layer compared with the mechanical method, i.e., the slitting method [27], and the sin^2^*ψ* method [24]. The second half demonstrates the residual stress measurement of the peened specimen by laser cavitation peening using the proposed 2D method.

## 2. Experimental Apparatus and Procedures

### 2.1. Peening Systems

To prepare specimens with compressive residual stress, cavitation peening (CP) using a cavitating jet (see Figure 2) and laser cavitation peening (LCP) using a pulse laser (see Figure 3) were applied. In the case of CP, a high-speed water jet was injected into a water-filled chamber. The cavitating conditions were the same as those in a previous paper [8]; the injection pressure of the jet was 30 MPa, the diameter of the nozzle *d* was 2 mm and the standoff distance was 222 mm. To enhance the peening intensity, the nozzle had a cavitator with a diameter *d_c_* of 3 mm [42] and an outlet bore with a diameter *D* and length *L* of 16 and 16 mm, respectively [43]. The specimen was installed in the recess. In the case of LCP, a Nd:YAG (Nd*:*Y_3_Al_5_O_12_) laser with Q-switch was used to generate laser cavitation [12]. The repetition frequency of the pulse laser was 10 Hz. The used wavelength was 1064 nm. The pulse laser was focused onto the specimen, which was placed in a water-filled glass chamber. The standoff distance in the air *s_a_* and in water *s_w_* was optimized by measuring the peening intensity [12]. The specimen based the stage was moved perpendicularly to the direction of the laser axis by the stepping motors.

As the backside surface of the peened plate had compressive residual stress [44], a recirculating shot peening (SP) system accelerated by a water jet [45] was used for the peening. Note that compressive residual stress was introduced onto the backside surface, but the grain size on the back side was not affected, as the backside surface was not peened. At SP, the shots were installed a chamber, whose diameter was 54 mm, and accelerated by the water jet through three holes with a diameter of 0.8 mm. The diameter and the number of the shots were 3.2 mm and 500, respectively. The injection pressure of the water jet was 12 MPa. The distance from the nozzle to the specimen surface was 50 mm. To avoid a loss of shots, the specimen was set in the recess.

### 2.2. Material

The tested material was austenitic stainless steel, Japanese Industrial Standards JIS SUS316L. Four different specimens were used, as shown in Table 1. All specimens were made from plates 2, 3 and 6 mm in thickness, respectively, and all the plates featured a No. 2B surface finish accomplished by temper rolling. Specimen A was used to measure the residual stress of the peened side by the 2D method compared with the slitting method and sin^2^*ψ* method. Specimens A, B and C were used to optimize the measuring conditions of the 2D method. Specimen D was used to demonstrate the effect of the laser spot on the residual stress distribution. During LCP, the surface was laser and the outermost surface showed tensile residual stress; the surface was then removed by electrochemical polishing. The peening intensity of CP, SP and LCP was controlled based on processing time per unit length, processing time and pulse density, respectively. The processing time per unit length of specimen A was chosen based on the values in the reference [8]. To introduce large compressive residual stress into the specimen, 100 pulses/mm^2^ pulse density and 6 mm thickness were chosen for specimen B. Considering the results of the preliminary experiment by measuring the residual stress, 30 s and 3 mm thickness were chosen for specimen C. For specimen D, the pulse density of 4 pulses/mm^2^ was optimized by measuring the fatigue life changes based on pulse density.

As mentioned above, the number of grains in the measurement area is important factor for the accuracy of the stresses using X-ray diffraction method. The average grain size, i.e., spatial diameter [46] and the grain size that occupied 50% of the area *d_N_* were measured. The *d_N_* was obtained by the following procedure. The area *A_i_* of each grain was measured and they were sorted from small value to large value, then the cumulated area *A_C_* was calculated as follows.
(1)$$A_{C}=\sum_{i=1}^{N}\;A_{i}$$
when the number *N* at *A_C_*/*A* = 50% was obtained, *d_N_* = $2\sqrt{A_{N}/\pi }$ was defined as the grain size that occupied 50%. Here, *A* was test area. 300 grains were measured in the present experiment.

Figure 4 illustrates a schematic diagram of specimen D and the coordinates of the residual stress with the scanning direction of the laser. The specimen was moved at 5 mm/s, and the reputation frequency of the laser was 10 Hz. Then, the specimen was stepped at 0.5 mm, as shown in Figure 4. The positional relationship of the laser spots in each row was different for each row as shown in Figure 4, as the stepping motors and pulse laser were not synchronized. Note that the rolling direction was *y*-direction in Figure 4.

### 2.3. Residual Stress Measurement

To confirm the compressive residual stress of specimen A introduced by cavitation peening, the residual stress was measured using the slitting method [27]. The slitting was done using a wire electric discharge machine (EDM, Sodick, Chicago, IL, USA), and the residual stress was evaluated from the strain obtained by the strain gage. The diameter of the wire was 0.254 mm and the gage length of the strain gage was 0.81 mm. The distribution of the residual stress with depth under the surface was obtained by using the recorded strain and solving an inverse problem following the procedure in the reference [15]. In the present paper, slitting of 0.0254 mm in depth was used as the reference value. The distribution of the residual stress with depth and more details on the slitting method can be found in the reference [8].

To measure the residual stress by X-ray diffraction, an XRD system (Bruker Japan K. K., Tokyo, Japan) with a two-dimensional position sensitive proportional counter (2D-PSPC) was used. The same system was used for the sin^2^*ψ* method [24] and 2D method [47]. The schematic diagram and coordinates *θ*, *ψ*, *ω*, *χ*, *φ* of the XRD system with 2D-PSPC are shown in Figure 5. The Kα X-rays from a Cr-tube operating at 35 kV and 40 mA were used. The used diffracted plane was γ-Fe (2 2 0), and the diffraction angle without strain was 128°. In the residual stress analysis for both the sin^2^*ψ* method and the 2D method, Relevant software (Leptos ver 7.9, Bruker Japan K. K., Tokyo, Japan) was used. The used Young’s modulus and Poisson ratio were 191.975 GPa and 0.3, respectively. To investigate the effects of the measuring area, five different collimators with diameters of 0.1460, 0.3, 0.5, 0.5724 and 0.8 mm were used. The 0.1460 and 0.5724 mm collimators were of the total reflection type, and the other collimators were of the double-holed type. Table 2 and Table 3 show the measuring conditions and analyzed areas of X-ray diffraction obtained using each method based on the standard method [24] and the previous report [26]. Under both the sin^2^*ψ* method and the 2D method, 24 frames were measured. In the case of the sin^2^*ψ* method, the X-ray diffraction profile obtained an accumulating X-ray diffraction of *χ* = 85–95°, and diffracted peaks 2*θ* were obtained at each *ψ*. Then, the residual stress was calculated from the sin^2^*ψ*−2*θ* diagram. To eliminate the *ψ* split, +*ψ* and −*ψ* were measured for each *x* and *y* direction. At the sin^2^*ψ* method, *σ_Rx_* was obained by *φ* = 90° and 270°, *σ_Ry_* was obtaied by *φ* = 0 and 180°, respectively. Namely, 12 frames in Table 2 were used to obtain *σ_Rx_* and *σ_Ry_*, respectively.

### 2.4. Observation of Specimen Surface

To investigate the grain size of the tested material, the surface was observed using a scanning electron microscope (SEM; JCM-7000, JEOL Ltd., Tokyo, Japan). The aspect of the laser-cavitation-peened specimen was also observed using a laser confocal microscope (VK-100, Keyence Corporation, Osaka, Japan) to obtain the surface profile.

## 3. Results

### 3.1. Comparison of Measured Residual Stress between the Slitting Method, sin^2^ψ Method and 2D Method

In order to compare the residual stress measured by the slitting method, the sin^2^*ψ* method and the 2D method, Figure 6 illustrates the residual stress *σ_Ry_* of specimen A. For the sin^2^*ψ* method and the 2D method, the effect of the measuring area was investigated by changing diameter of the collimator *d_col_*. As shown in Figure 6, the exposure time at each frame *t_exp_* was also changed based on the area of the collimator. In the case of *d_col_* = 0.8 mm and *t_exp_* = 20 s, the specimen was moved in both directions, *x* = ±2 mm and *y* = ±2 mm, to minimize the exposure time. As shown in Figure 6, in the case of *d_col_* = 0.8 mm and *t_exp_* = 40 s at *x* = ±0 mm and *y* = ±0 mm, the residual stress *σ_R_* measured using the sin^2^*ψ* method and the 2D method was −220 ± 74 and −220 ± 14 MPa; these results are similar to −251 ± 16 MPa, which was measured by the slitting method. For the sin^2^*ψ* method, the residual stress measured using *d_col_* ≥ 0.5724 mm was similar to that of the slitting method. However, at *d_col_* ≤ 0.5 mm, the residual stress was too small and the standard deviation of the residual stress was too large. For the reference, Appendix A reveals the diagram of sin^2^*ψ* − 2*θ* for *d_col_* = 0.146 mm, *t_exp_* = 20 min and *d_col_* = 0.8 mm, *t_exp_* = 40 s. On the other hand, in the case of the 2D method, the residual stress measured using *d_col_* = 0.146 mm was −187 ± 29 MPa. Thus, it can be concluded that the 2D method can evaluate the residual stress by using *d_col_* = 0.146 mm. Specifically, the 2D method can measure the residual stress in the 1/15 region of the sin^2^*ψ* method under the presented conditions.

In order to investigate the difference in the measurement of residual stress between the sin^2^*ψ* method and the 2D method for the austenitic stainless steel tested using a collimator of *d_col_* = 0.146 mm, Figure 7 shows the aspects of the surface of the measured sample observed using a scanning electron microscope (SEM). The average grain size, i.e., spatial diameter [46], was 6.6 ± 4.0 μm in diameter, and the grain size that occupied 50% of the area was about 11 μm. As shown in Figure 7, specific anisotropy is not observed. Thus, at the present condition, the rolling direction did not affect the results of residual stress measurement.

Figure 8 illustrates a typical X-ray diffraction pattern that was a part of the Debye ring, as detected by 2D-PSPC from the stainless steel treated by cavitation peening—i.e., specimen A treated using *d_col_* = 0.146 mm—and the analyzed area for (a) the sin^2^*ψ* method and (b) the 2D method. As illustrated in Figure 7, the grain size was about 1/10 of the diameter of the collimator, and the X-ray diffraction pattern was a mottled pattern, as shown in Figure 8. In the case of the sin^2^*ψ* method, the diffraction pattern located near *χ* ≈ 90° should be used due to the theory of the sin^2^*ψ* method; the standard deviations of the sin^2^*ψ* method at *d_col_* ≤ 0.5 mm were remarkably large, as the diffraction pattern at *χ* ≈ 90° was weak or not obtained. For the present residual stress analysis, *χ* = 85–95° was used for the sin^2^*ψ* method. The residual stress obtained by the sin^2^*ψ* method for *d_col_* = 0.8 mm, *t_exp_* = 40 s, *x* = 0 mm and *y* = 0 mm was −300 ± 46 MPa based on analysis using 2*θ* = 125–132° and *χ* = 70–110°. These values were too large compared to the value of −251 ± 16 MPa measured by the slitting method. Namely, when large area, i.e., *χ* = 70–110°. was used, the number of counts of the X-ray diffraction was increased. However, *χ* = 70–110° was too large for the sin^2^*ψ* method. On the other hand, *χ* = 70–110° was used for the 2D method, as the 2D method obtained the residual stress from the distortion of the Debye ring. For the 2D method, the Debye ring was obtained by interpolation and extrapolation of the patchy patterns of the diffraction spots. It was concluded that the 2D method could evaluate the residual stress in a smaller area compared to the sin^2^*ψ* method, as the Debye ring was used in the relatively large area of *χ*. Thus, it can be said that the key point for evaluating the residual stress in a small area is to obtain a more uniform Debye ring.

### 3.2. Optimum Condition for the 2D Method to Evaluate Residual Stress

In the present paper, to obtain a better Debye ring, specimen oscillation in the *ω* direction—i.e., *ω*-oscillation—was proposed. Figure 9 shows the X-ray diffraction pattern (a) without *ω*-oscillation (i.e., ∆*ω* = 0°) and (b) with *ω*-oscillation at ∆*ω* = 10°. As shown in Figure 9a (*ψ* = 0, *φ* = 0) and (b) (*ψ* = 0, *φ* = 0), the diffraction spot at *χ* ≈ 102° became a streak by *ω*-oscillation. Precisely, the diffraction spot became a streak in the *χ* direction by *ω*-oscillation of the specimen. The *ω*-oscillation helped to achieve a better Debye ring. Note that, in the case of 2D method, there should be an optimum value of ∆*ω*, as the residual stress was obtained by the distortion of Debye ring. The optimum value of ∆*ω* is discussed in the following.

In order to investigate the *ω*-oscillation of the specimen both qualitatively and quantitatively, Figure 10 shows the typical X-ray diffraction pattern of specimen A at *ψ* = 0° and *φ* = 0 changing with (a) the collimator diameter *d_col_*, (b) the exposure time *t_exp_* and (c) the *ω*-oscillation angle ∆*ω*. Figure 11 illustrates the relationship between the total count of X-ray diffraction and the standard deviation of the residual stress ∆*σ_R_*. As shown in Figure 10a, the X-ray diffraction pattern became a mottled pattern with a decrease in the diameter of the collimator *d_col_*. Then, the ∆*σ_R_* increased with a decrease of *d_col,_* as shown in Figure 11. When the exposure time *t_exp_* was increased, the Debye ring became clear, as shown in Figure 10b, and then ∆*σ_R_* decreased with an increase of *t_exp_*. As shown in Figure 11, ∆*σ_R_* decreased with an increase in the total count of X-ray diffraction for both *d_col_* and *t_exp_*. As shown in Figure 10c, the diffraction pattern changed from a mottled pattern to a streak-like pattern with an increase of ∆*ω*. The ∆*σ_R_* was 64 MPa for ∆*ω* = 0°, 49 MPa for ∆*ω* = 2°, 49 MPa for ∆*ω* = 4°, 37 MPa for ∆*ω* = 6°, 29 MPa for ∆*ω* = 8° and 32 MPa for ∆*ω* = 10°. Specifically, ∆*σ_R_* decreased with an increase of ∆*ω* at ∆*ω* = 0–8° and presented its minimum at ∆*ω* = 8°; then, ∆*σ_R_* slightly increased at ∆*ω* = 10°. As the 2D method evaluates the residual stress caused by distortion of the Debye ring, a ∆*ω* that is too large dims the distortion by averaging too large an area in the *χ* direction. Thus, it can be concluded that the *ω*-oscillation of the specimen was effective, and the optimum value of ∆*ω* was 8°. In Figure 11, ∆*σ_R_* = 64 MPa. In Figure 11, ∆*σ_R_* = 64 MPa for ∆*ω* = 0° corresponds to *t_exp_* = 4 or 5 min and ∆*σ_R_* = 29 MPa for ∆*ω* = 8° corresponds to *t_exp_* = 20 min. Thus, *ω*-oscillation of the specimen has the effect of shortening the measurement time to 1/4–1/5.

In order to confirm that the 2D method can evaluate the residual stress of the austenitic stainless steel by using the 0.146 mm-diameter collimator, Figure 12a reveals the residual stress (*σ_Rx_*, *σ_Ry_)* of specimen A, B and C as a function of the diameter of the collimator *d_col_*, and Figure 12b shows the standard deviation of the residual stress. The data for specimen A in Figure 6 were used as the *σ_Ry_* in Figure 12. For all three specimens, A, B and C, as both *σ_Rx_* and *σ_Ry_* at *d_col_* = 0.146 mm were nearly equal to the values at *d_col_* = 0.8 mm, the residual stresses of *d_col_* = 0.146 mm for specimens A, B and C were able to be evaluated.

In the case of specimen B, i.e., LCP, the specimen was treated with 100 pulses/mm^2^, as mentioned in Table 1. The specimen was moved in the *x*-direction at 1 mm/s. As the repetition frequency of the pulse laser was 10 Hz, the distance of each laser spot was 0.1 mm. After each specimen was treated in the *x*-direction, it was moved stepwise at 0.1 mm in the *y*-direction. As shown in Figure 12a, *σ_Rx_* and *σ_Ry_* were about −400 MPa and −670 MPa, respectively. Specifically, the compressive residual stress introduced in *y*-direction, i.e., the stepwise direction, under laser cavitation peening was 270 MPa larger than that in the *x*-direction. Even though the distances between the laser spots in both *x*- and *y*-direction were the same, the compressive residual stress introduced in the *y*-direction was larger than that in the *x*-direction. This tendency was similar to the results in the reference [38].

In the present experiment, SP was applied, then the residual stress on the back side of shot peened specimen was measured to avoid the effects of the change of the grain size etc. If the treated surfaces by SP, CP and LCP were evaluated, the characteristics of the peened surfaces have different features. It was reported that the dislocation density of CP and LCP of austenitic stainless steel SUS316L was lower than that of SP at the equivalent peening condition, i.e., the equivalent arc height condition [48].

In order to investigate the effects of *ω*-oscillation of the specimen, Figure 13 reveals the residual stress (*σ_Rx_*, *σ_Ry_*) and the standard deviation of the residual stress ∆*σ_R_* as a function of the *ω*-oscillation angle ∆*ω* for specimens A, B and C. As shown in Figure 13, the residual stress of specimen A, B and C was roughly constant for all ∆*ω* values, and ∆*σ_R_* roughly decreased with an increase of ∆*ω*. In the case of LCP, i.e., specimen C, the compressive residual stress increased with an increase of ∆*ω* at ∆*ω* ≤ 8°; then, the compressive residual stress slightly decreased at ∆*ω* = 10°. Further, the ∆*σ_R_* of specimen C had its minimum at ∆*ω* = 8°, and the ∆*σ_R_* at ∆*ω* = 10° was larger than the ∆*σ_R_* of ∆*ω* = 8°. As the 2D method obtained the residual stress from the distortion of the Debye ring, a ∆*ω* value that was too large caused a decrease of the residual stress and an increase of ∆*σ_R_*, just as with specimen A. It can be concluded that the *ω*-oscillation of the specimen is effective for evaluating the residual stress and that the optimum value of ∆*ω* is 8°.

In order to determine the optimum exposure time needed to obtain the X-ray diffraction pattern, Figure 14 shows (a) the residual stress *σ_Rx_*, *σ_Ry_* and (b) standard deviation of the residual stress ∆*σ_R_* as a function of exposure time *t_exp_* for specimens A, B and C. Under all measurement conditions in Figure 14, the specimens were oscillated at ∆*ω* = 8°, and the diameter of the collimator was 0.146 mm. For specimens A and B, the *σ_Rx_* and *σ_Ry_* were nearly constant at all *t_exp_* values. For specimen C, *σ_Rx_* and *σ_Ry_* decreased and became saturated at *t_exp_* = 15 and 20 min. The ∆*σ_R_* of specimens A, B and C decreased with an increase of *t_exp_* and became saturated at *t_exp_* = 15 or 20 min. Under the studied conditions, *t_exp_* = 20 min is the optimum exposure time to obtain residual stress. The X-ray diffraction totaled about 4.6 × 10^4^ counts.

### 3.3. Residual Stress Distribution of Specimen Treated by Laser Cavitation Peening

To determine the typical results for the residual stress of austenitic stainless steel JIS SUS316L in the local area measured by the 2D method, the residual stress of the specimen treated by laser cavitation peening, i.e., that of specimen D, was evaluated using the 2D method. Considering the results in Section 3.1 and Section 3.2, the measuring conditions were as follows: The diameter of the collimator *d_col_* was 0.146 mm, the *ω*-oscillation angle ∆*ω* was 8° and the exposure time of each frame *t_exp_* was 20 min. The pulse density of the laser cavitation peening *d_L_* was 4 pulses/mm^2^, as the fatigue life was greatest at *d_L_* = 4 pulses/mm^2^ and changed with the pulse density [23]. Under these conditions, the laser spot distances in the *x*- and *y*-directions were 0.5 and 0.5 mm, respectively. As mentioned above, the top surface revealed tensile residual stress, and then the surface of 33 μm was removed by electrochemical polishing. Note that the residual stress at 30 μm accurately corresponded to the fatigue life, as the fatigue life estimated by the residual stress at 30 μm, the surface hardness and the surface roughness was proportional to the experimental value [23].

Figure 15 shows (a) the aspects of the laser-cavitation-peened specimen observed using a CCD camera on the XRD system and (b) the aspects of the same specimen observed using a laser confocal microscope. As the specimen was treated with *d_L_* = 4 pulses/mm^2^, the distances of the *x*- and *y*-directions between the laser spots were 0.5 and 0.5 mm. Under the presented conditions, the depth of the laser spot was about 10 μm. As shown in Figure 15, the laser spot diameter was about 0.5 mm after the 33 μm electrochemical polishing. The vertical positional relationship of the laser spots was not aligned, as the stepping motor and laser system were not synchronized.

Figure 16 shows the residual stress distribution as a function of *y* at *x* = 0, 0.125, 0.25, 0.375 and 0.5 mm for (a) *σ_Rx_* and (b) *σ_Ry_*. The standard deviation at the residual stress was about 70 MPa. It was difficult to recognize the 0.5 mm interval period at each position of *x*, as shown in Figure 16a,b, because the positional relationships of the laser spots at *y* = 0, 0.5, 1, 1.5 and 2 mm were different, as shown in Figure 15. As *σ_Rx_* and *σ_Ry_* varied from 0 to −150 MPa, there was a difference of about 150 MPa depending on the location for both *σ_Rx_* and *σ_Ry_*.

Figure 17 reveals the residual stress *σ_Rx_* and *σ_Ry_* changing with distance *x* at *y* = 0 with the laser spot aligned in the *x*-direction. The *σ_Rx_* was about −100 MPa at *x* = 0 mm; it had a peak of 0 MPa at *x* = 0.125 mm and then decreased with an increase in *x*. Then, *σ_Rx_* had a minimum of 150 MPa at *x* = 0.375 mm and increased to 0 MPa at x = 0.5 mm. On the other hand, *σ_Ry_* had a minimum at x = 0.125 mm and a maximum at x = 0.375 mm. Even though the standard deviations were somewhat large, a 0.5 mm cycle was observed for both *σ_Rx_* and *σ_Ry_*. It can be concluded from Figure 16 and Figure 17 that the residual stress may differ by about 150 MPa depending on the location when austenitic stainless steel JIS SUS316L is treated using laser cavitation peening at 4 pulses/mm^2^. As shown in Figure 12 and Figure 13, when the residual stress was relatively uniform, the experimental deviation at the present condition using the 0.146 mm collimator was about ±40 MPa. At the measurement of LCP specimen treated at 4 pulse/mm^2^, the residual stress was changed from 0 MPa to −150 MPa within 0.25 mm in length, thus the experimental deviation using the 0.146 mm collimator was about ±70 MPa due to the spatial distribution.

## 4. Conclusions

To clarify the possibilities and measure the conditions of residual stress in a mechanical-surface-modified layer with a small area by the 2D method using X-ray diffraction, the residual stress of the austenitic stainless steel JIS SUS316L treated by cavitation peening was measured by the 2D method changing with the diameter of the collimator comparing with the sin^2^*ψ* method and the slitting method. The measured sample was austenitic stainless steel with temper rolling. The average diameter and 50% area of the grain size of the tested specimen were 6.6 ± 4.0 μm and about 11 μm, respectively. The specimens were treated by cavitation peening using a cavitating jet and a pulse laser, i.e., laser cavitation peening. The results obtained can be summarized as follows:(1)Compared to the sin^2^*ψ* method, the 2D method can evaluate the residual stress in a small area, which is 1/15 of the area ratio of the sin^2^*ψ* method. In the present experiment, the measurable areas of the sin^2^*ψ* method and 2D method were 0.5724 mm in diameter and 0.146 mm in diameter, respectively.(2)The *ω*-oscillation of the specimen using the 2D method had the effect of reducing the measurement error to 1/2. This result is equivalent to the effect of reducing the measurement time to 1/5–1/4. The optimum *ω*-oscillation angle ∆*ω* was 8°.(3)The 2D method using optimized conditions can evaluate the residual stress distribution for a laser spot with a diameter of 0.5 mm.(4)The compressive residual stress under laser cavitation peening at 100 pulses/mm^2^ was larger in the stepwise direction than in the orthogonal direction.

## Figures and Tables

**Figure 1 materials-14-02772-f001:**
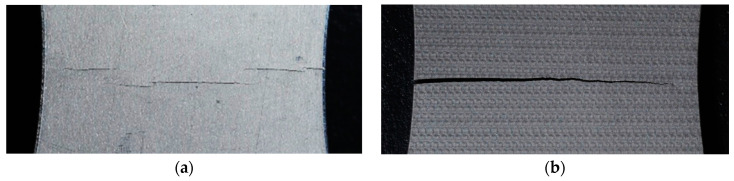
Aspect of a fractured fatigue specimen of stainless steel. (**a**) Non-peened specimen (*σ_a_* = 301 MPa, *N_f_* = 7.8 × 10^5^); (**b**) specimen treated by laser cavitation peening (*σ_a_* = 308 MPa, *N_f_* = 4.8 × 10^6^).

**Figure 2 materials-14-02772-f002:**
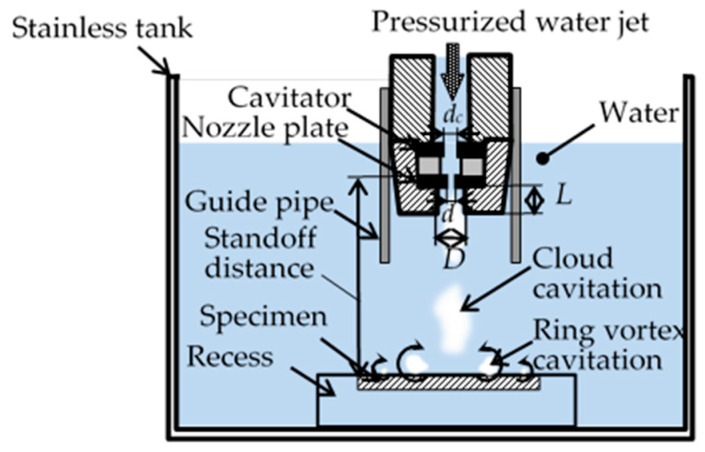
Schematic diagram of cavitation peening (CP) using a cavitating jet.

**Figure 3 materials-14-02772-f003:**
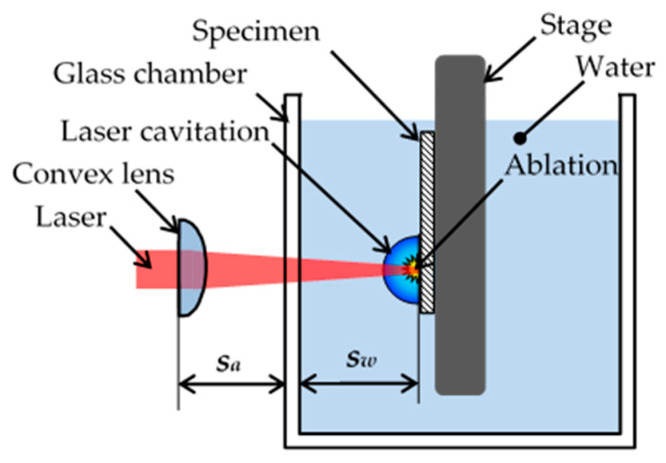
Schematic diagram of cavitation peening using a pulse laser, i.e., laser cavitation peening (LCP).

**Figure 4 materials-14-02772-f004:**
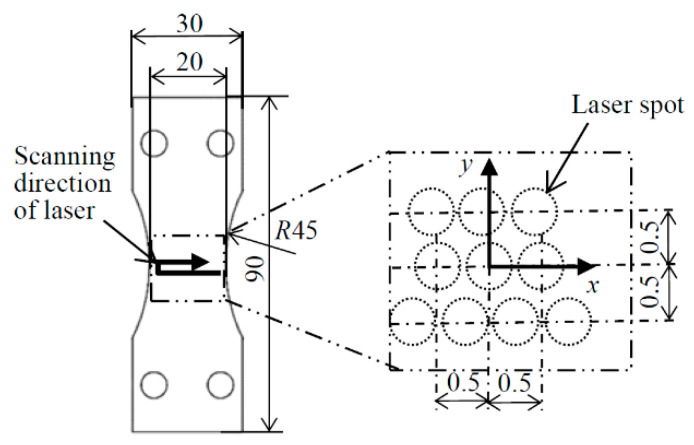
Schematic diagram of specimen D treated by laser cavitation peening and the coordinates for residual stress measurement.

**Figure 5 materials-14-02772-f005:**
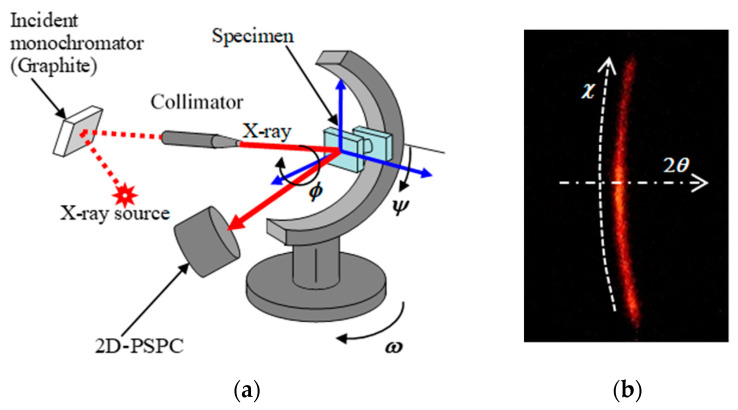
Schematic diagram and coordinates of XRD system with 2D-PSPC. (**a**) Coordinates of the system; (**b**) coordinates on the 2D-PSPC.

**Figure 6 materials-14-02772-f006:**
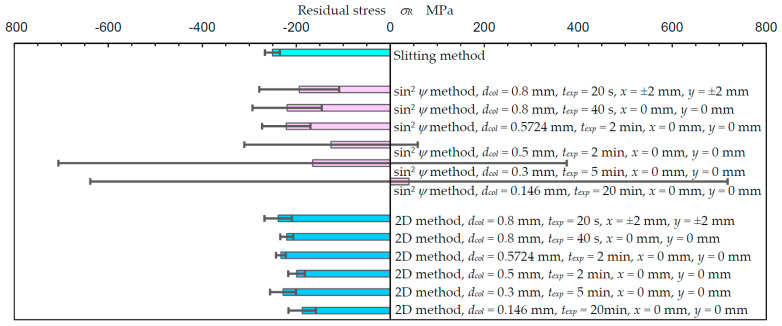
Comparison of the slitting method, sin^2^*ψ* method, and the 2D method for the residual stress of stainless steel treated by cavitation peening (specimen A).

**Figure 7 materials-14-02772-f007:**
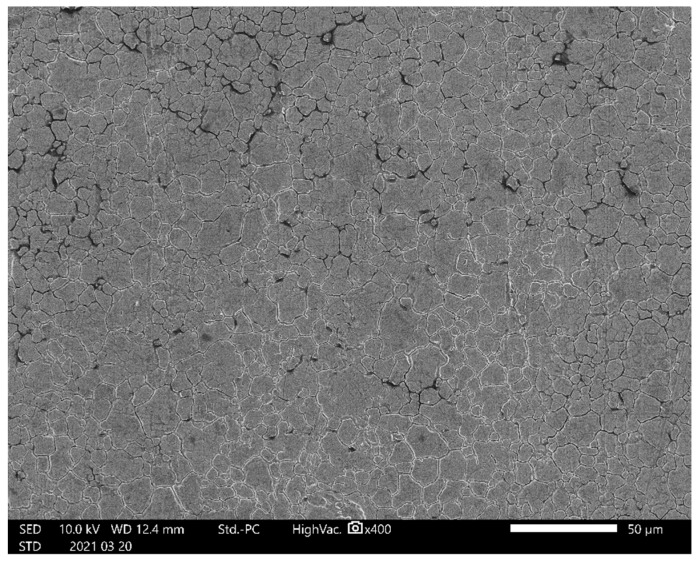
Aspect of the surface of specimen A observed by scanning electron microscope (SEM).

**Figure 8 materials-14-02772-f008:**
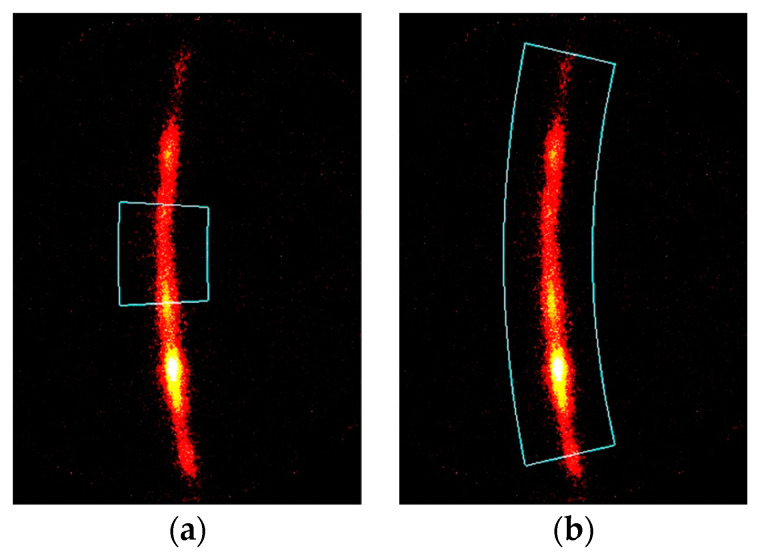
Typical X-ray diffraction pattern detected by 2D-PSPC from stainless steel treated using cavitation peening (Specimen A) and the analyzed area (*d_col_* = 0.146 mm, *ψ* = 0°, *φ = 0*°, ∆*ω* = 0°, *t_exp_* = 20 min, *x* = 0 mm, *y* = 0 mm); (**a**) analyzed area for the sin^2^*ψ* method (2*θ* = 125–132°, *χ* = 85–95°); (**b**) analyzed area for the 2D method (2*θ* = 125–132°, *χ* = 70–110°).

**Figure 9 materials-14-02772-f009:**
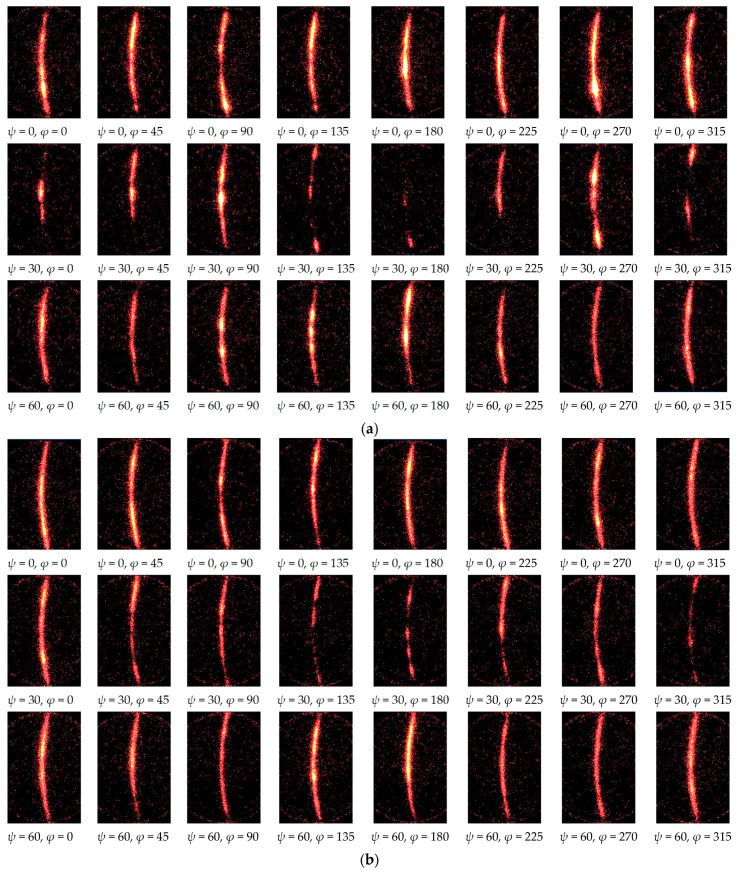
X-ray diffraction patterns detected by 2D-PSPC from the stainless steel treated by cavitation peening (Specimen A). (**a**) ∆*ω* = 0°; (**b**) ∆*ω* = 10°.

**Figure 10 materials-14-02772-f010:**
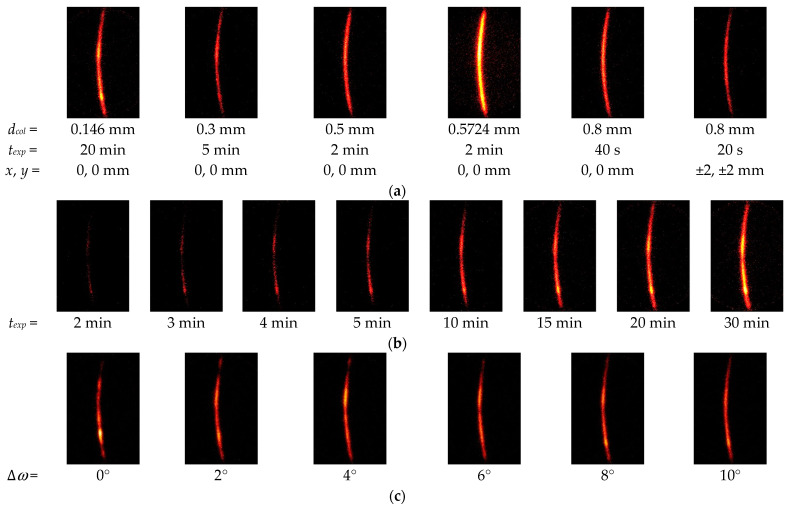
Typical X-ray diffraction patterns detected by 2D-PSPC at *ψ* = 0° and *φ* = 0° from stainless steel treated by cavitation peening (Specimen A). (**a**) effect of the collimator diameter *d_col_* at ∆*ω* = 8°; (**b**) effect of exposure time *t_exp_* at *d_col_* = 0.146 mm and ∆*ω* = 8°; (**c**) effect of the *ω*-oscillation angle ∆*ω* at *d_col_* = 0.146 mm and *t_exp_* = 20 min.

**Figure 11 materials-14-02772-f011:**
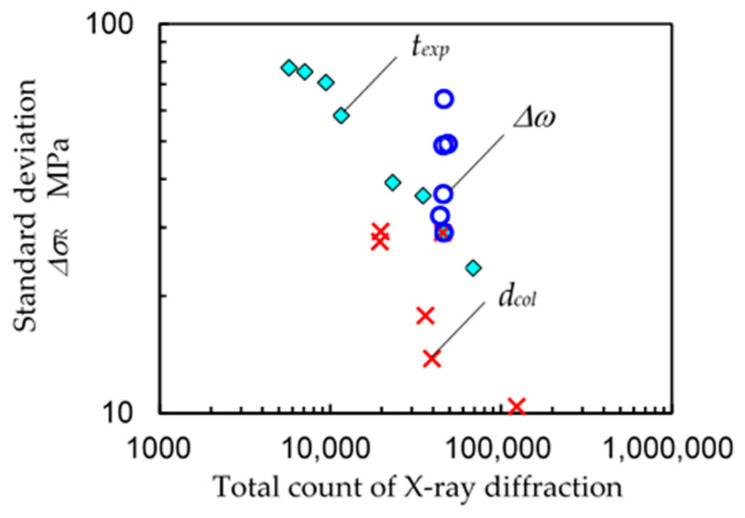
Relationship between the counts at *ψ* = 0° and *φ* = 0° detected by 2D-PSPC and the standard deviation of the residual stress of stainless steel treated by cavitation peening (specimen A) changing with exposure time *t_exp_*, collimator diameter *d_col_* and *ω*-oscillation angle ∆*ω*.

**Figure 12 materials-14-02772-f012:**
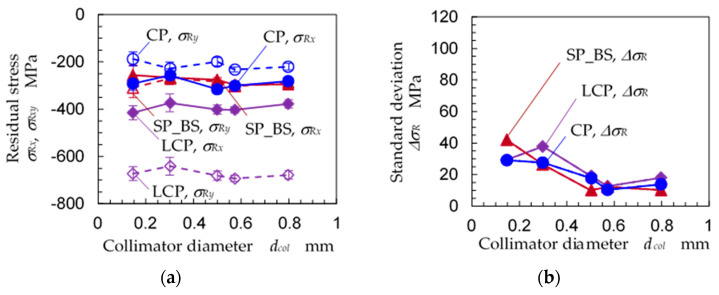
Effect of the collimator dimeter on the residual stress measurement of the stainless steel by the 2D method (∆*ω* = 8°); CP: specimen A, LCP: specimen B, SP_BS: specimen C. (**a**) residual stress; (**b**) standard deviation of residual stress.

**Figure 13 materials-14-02772-f013:**
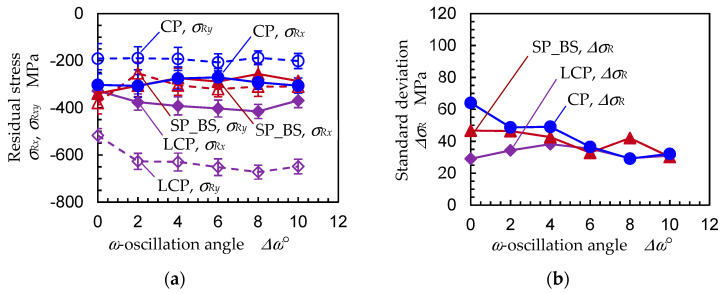
Effect of the *ω*-oscillation angle ∆*ω* on the residual stress measurement of stainless steel using the 2D method (*d_col_* = 0.146 mm, *t_exp_* = 20 min); CP: specimen A, LCP: specimen B, SP_BS: specimen C; (**a**) residual stress; (**b**) standard deviation of residual stress.

**Figure 14 materials-14-02772-f014:**
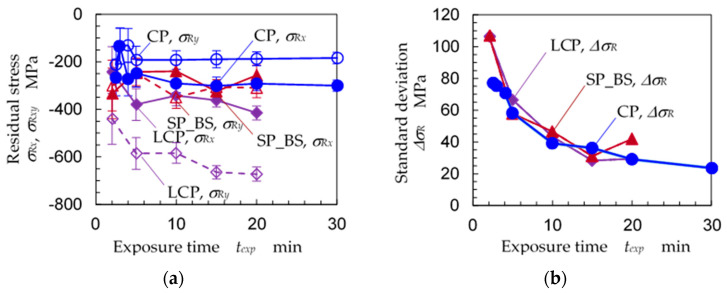
Effect of exposure time *t_exp_* on the residual stress measurements of stainless steel under the 2D method (*d_col_* = 0.146 mm, ∆*ω* = 8°); CP: specimen A, LCP: specimen B, SP_BS: specimen C. (**a**) Residual stress; (**b**) standard deviation of residual stress.

**Figure 15 materials-14-02772-f015:**
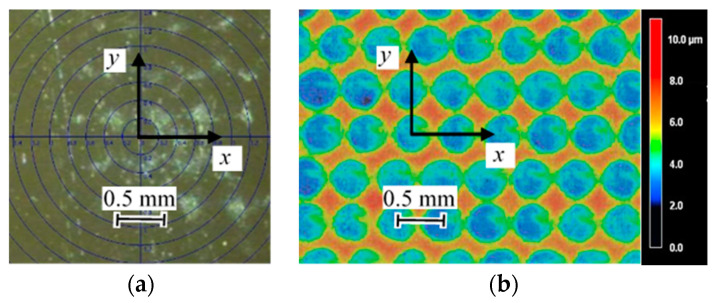
Aspects of the surface of the stainless-steel specimen treated by laser cavitation peening (specimen D); (**a**) observation using the CCD camera on the XRD system; (**b**) observations from the laser confocal microscope.

**Figure 16 materials-14-02772-f016:**
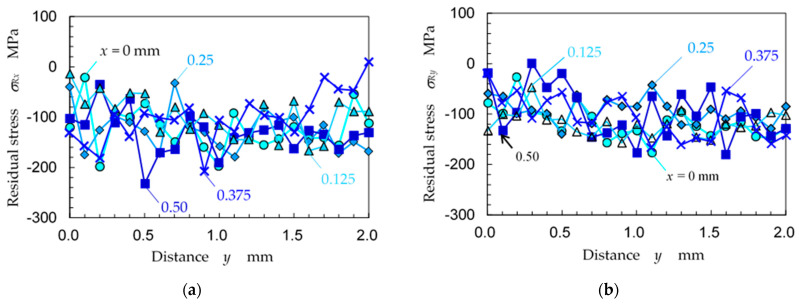
Distribution of the residual stress changing with distance *y* at various positions of *x* (specimen D); (**a**) residual stress in the *x*-direction *σ_Rx_*; (**b**) residual stress in the *y*-direction *σ_Ry_*.

**Figure 17 materials-14-02772-f017:**
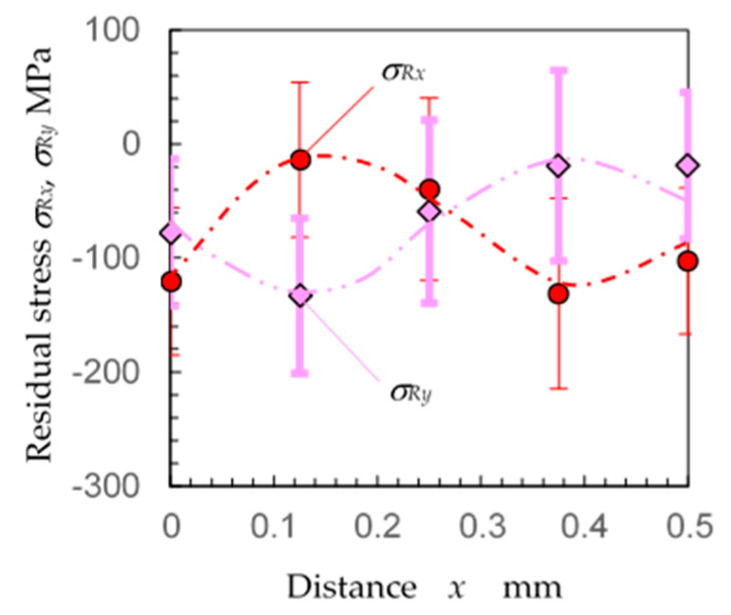
Distribution of the residual stress *σ_Rx_* and *σ_Ry_* changing with distance *x* (specimen D).

**Table 1 materials-14-02772-t001:** Specimens for residual stress measurements.

Symbol	Peening Method	Peening Intensity	Thickness	Measured Side	Electrochemical Polishing
A	Cavitation peening CP	8 s/mm	2 mm	Peened side	None
B	Laser cavitation peening LCP	100 pulses/mm^2^	6 mm	Peened side	39 μm
C	Shot peening SP	30 s	3 mm	Back side	None
D	Laser cavitation peening LCP	4 pulses/mm^2^	2 mm	Peened side	33 μm

**Table 2 materials-14-02772-t002:** Measuring conditions for each method.

Method	*ψ*°	*φ*°
sin^2^*ψ* method	0	0, 90, 180, 270
	20.268	0, 90, 180, 270
	29.334	0, 90, 180, 270
	36.870	0, 90, 180, 270
	43.854	0, 90, 180, 270
	50.768	0, 90, 180, 270
2D method	0	0, 45, 90, 135, 180, 225, 270, 315
	30	0, 45, 90, 135, 180, 225, 270, 315
	60	0, 45, 90, 135, 180, 225, 270, 315

**Table 3 materials-14-02772-t003:** Analyzed area of obtained X-ray diffraction.

Method	2*θ*°	*χ*°
sin^2^*ψ* method	125–132	85–95
2D method	125–132	70–110

## Data Availability

The data presented in this study are available on request from the author.

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
