# Peer review of "Optimization of Residual Stress Measurement Conditions for a 2D Method Using X-ray Diffraction and Its Application for Stainless Steel Treated by Laser Cavitation Peening"

_materials, 2021, doi:10.3390/ma14112772_

Round 1

Reviewer 1 Report

Could be a bit clearer why we need a method that can measure residual stress at a sub- millimeter level.
Why isnt the mm scale enough if there is enough overlay?

Comments for authors:

Authors present a research study dealing with residual stress measurements employing three different methods (slitting, sin2 and 2D detector) carried out on surfaces subjected to cavitation and shot peening. The study provides an original contribution showing some advantages of the 2D method regarding measurement resolution on specimens exposed to laser cavitation. Nevertheless, some issues should be revised in the manuscript.  

Title:

Introduction

- Slitting method fundamentals are not mentioned in the introduction. A short explanation on this method would be useful for readers.

- In light of the relevance of the w-oscillation parameter, authors should include a concise explanation regarding the fundamentals of this topic in connection to the 2D method. 

Experimental

- Basic details of the shot peening procedure should be shown in the paper.

Discussion, results and conclusions.

- It is expected that shot peening and cavitation peening produce residual stresses with different features (i.g possible orientation dependence). Please include this topic in the discussion.

- Figs 16 and 17: The experimental deviations seem to be moderately high. Although the transition zones between cavitation impacts are detectable according to Fig. 17, authors should analyse the precision of the technique in depth. Moreover, a realistic conclusion regarding precision and applicability of the 2D method should be given.   

Minor points:

- Fig. 1. Were these results previously published? The reference of the study is needed.

- Abbreviation for "Laser cavitation peening" is explained in page 3 and then again in page 4

- Typos: Page 1: L13 (“methods)

Reviewer 2 Report

Dear Authors,

The paper combines two methods of the residual stress mesuments. The goal of the research seems worth investigation. I have the following quesions or comments on the manuscirpt. 

General question.

You invesitgated  staninless steel grade 316 and the samples were probably machined from the rolled plate. 

Did you taken into account the rolling direction? Did the rolling direcion can affect the results of stresses mesument (sample could be positioned in XRD system along rolling direction or perpendicular to rolling direction).

Table 1 shows that different thickness were invesitgated. Dont you think that the smaples with different level of strain have different level of stressess and it could affect the reuslts even you eatch the surface layer? 

Specific comments:

Authors wrote that: 

As the size of 37
the measured area using X-ray diffraction is similar to that of the distribution of the resid- 38
ual stress of the welding part and/or the laser-peened surface, it is necessary to improve 39
the accuracy of residual stress measurements by using X-ray diffraction

Gnenerally this phrase is blured. Please explain and clearly explain why the size is simillar to the distribution of the resitual stresses?

Please explain what factors affect the stersses measument accurancy using XRD?

You wrote "

by laser peening without protective coating [21]

" - provide details regarding the coating.

Please explain why the fig.1 fracture shape of peened and unpeend sample differs? 

Where samples positioned in XRD system in rolling direction or perpendicular?
You wrote that average grain size was 6.6 +/- 4.4 and the 50% of grains was about 11 um. Did you measured the grain size in the direction of the rolling or perpendicular.

If max grain size is: 6.6+4.4 =11um; how is it possible that the 50% of grains was about 11um? Please provide some more results regarding the grain size to justified obtained results or reconsider your calculations. 

Please explain the following phrase: 

The residual stress at 30 m accurately corre- 331
sponded to the fatigue life [23]

In the conclusions after "austenitic stainless steel" add the grade of your steel. 

Reviewer 3 Report

Dear authors, 
thank you very much for the article. 
I didn't find any major problems.

I have some notes/suggestions/recommendations:

  • unify the font size, add non-breaking spaces (e.g. line 134: 5 μm), re-check the use of long hyphens (e.g. line 61), use "Figure" instead of "Fig."
  • Line 13: ...is a common method...
  • Line 14: ...a method using a 2D...
  • Line 46: Using sin2psi method, the triaxial stress state is possible to calculated too without a multi-axis goniometer necessity.
  • Line 58: ...with depth was...
  • Line 80: The sentence "However..." should be deleted.
  • Line 122: ...ablated by laser ablation? Sounds better "laser ablated".
  • Line 160: ...Poisson ratio... 
  • Line 171: SIGMA_Ry, the whole sentence sounds strange.
  • Figure 5: Collimator
  • Table 3: You presented Chi=65-115deg, but there is written "70-110" in the rest of the cases.
  • Line 216: There is "x" instead of "Chi".
  • Line 217: I didn't understand that sentence. Moreover, the value -300 MPa is not mentioned in Fig.6.
  • According to my opinion, Fig. 9 contains unnecessarily many figures, 1/3-1/4 should be enough.
  • Line 265: I think, the "Delta omega" is missing.
  • Line 269: In the context of the article, the information about titanium alloy is superfluous.
  • Figs. 12, 13, 14 are insufficiently readable.
  • Fig. 16: Is it necessary? The text is sufficient.
  • Lines 376-8: Use "omega" sign.

I have some questions for authors:

  1. According to Tab. 2, you calculated four RS using sin2psi method and seven for 2D method. But, there is only one value in Fig. 6. What value is presented in Fig. 6 - Sigma_R; any average, or Sigma_Rx?
  2. How looks like some theta(sin2psi) plots, e.g. sin2psi method, d=0.146mm case from Fig. 6?
  3. Why you use macroscopic elastic constants? For XRD, the  X-ray elastic constants are much more suitable. Moreover, please correct the Youngs modulus units.

Round 2

Reviewer 2 Report

Dear Authors,

Thank you for your responses - I accept them all. 

Good luck in your future works!

With regards,

#Rev.

Author Response

Dear Reviewer,

Thank you for your response. 

With best regards,

Hitoshi Soyama